# Drug delivery from a solid formulation during breastfeeding—A feasibility study with mothers and infants

**Theresa Maier[1,2], Paula Peirce[3], Laura Baird[3], Sophie L. Whitehouse[4], Nigel K. H. Slater[1], Kathryn Beardsall[2,3] ***

**1** Department of Chemical Engineering and Biotechnology, University of Cambridge, West Cambridge Site, Cambridge, United Kingdom, **2** Department of Paediatrics, University of Cambridge, Cambridge, United Kingdom, **3** Cambridge University Hospitals NHS Foundation Trust, Cambridge, United Kingdom, **4** College of Social Sciences, Arts and Humanities, University of Leicester, Leicester, United Kingdom

\* kb274@medschl.cam.ac.uk, kb274@cam.ac.uk

**Data Availability Statement:** All relevant data are within the manuscript and its Supporting Information files.

## Abstract

### Background

Breastfeeding is critical to health outcomes, particularly in low-resource settings where there is little access to clean water. For infants in their first twelve months of life, the delivery of medications is challenging, and use of oral syringes to deliver liquid formulations can pose both practical and emotional challenges.

### Objective

To explore the potential to deliver medicine to infants via a solid formulation during breastfeeding.

### Methods

Single center feasibility study within a tertiary level neonatal unit in the UK, involving twenty-six breastfeeding mother-infant dyads. A solid formulation of Vitamin B12 was delivered to infants during breastfeeding. Outcomes included the quantitative change in serum vitamin B12 and assessment of maternal expectations and experiences.

### Results

Delivery of Vitamin B12 through a solid formulation that dissolved in human milk did not impair breastfeeding, and Vitamin B12 levels rose in all infants from a mean baseline (range) 533 pg/mL (236–925 pg/mL) to 1871 pg/mL (610–4981 pg/mL) at 6–8 hours post-delivery. Mothers described the surprising ease of 'drug' delivery, with 85% reporting a preference over the use of syringes.

### Conclusions

Solid drug formulations can be delivered during breastfeeding and were preferred by mothers over the delivery of liquid formulations via a syringe.

**Funding:** University of Cambridge WD Armstrong Trust (no grant number [to TM]), https://www.tech.cam.ac.uk/wd-armstrong-trust-fund University of Cambridge Kurt Hahn Trust (no grant number [to TM]), https://www.iso.admin.cam.ac.uk/grants-and-scholarships/kurt-hahn-trust/awards-german-students German Academic Scholarship Foundation (no grant number [to TM]), https://www.studienstiftung.de/en/ Applicable to all funders above: The funders had no role in study design, data collection and analysis, decision to publish, or preparation of the manuscript.

**Competing interests:** The authors have declared that no competing interests exist.

## Introduction

Breastfeeding saves lives, improves health and cuts health costs worldwide. In low-resource settings the delivery of medications, such as de-worming and retrovirals as well as nutrient and mineral supplementation, in liquid formulations is challenging, as liquids need to be refrigerated for storage, and syringes require clean water for sterilization [1]. However recent studies have also highlighted the emotional and practical challenges for many parents, of treating infants with liquid formulations as many infants demonstrate aversive behavior towards drug delivery from oral syringes [2]. The potential alternative of solid formulation drug delivery during breastfeeding was previously shown during *in-vitro* (lab-based) studies using a breastfeeding simulation apparatus [3, 4]. Now, this study aimed to evaluate the feasibility and acceptability of solid formulation delivery in a clinical setting with recruited infants while breastfeeding. It is the first clinical feasibility study of solid formulation delivery during breastfeeding to be undertaken with mother-infant dyads. Vitamin B12 was chosen as a therapeutic, due to its physiological importance for infant development, and its known safety profile over a wide dosage range [5]. It served as a model compound for other possible therapeutic formulations.

## Methods

### Study design

This was a single center feasibility study, at the University of Cambridge Addenbrooke's Hospital Trust (study start date: 9 July 2018, study end date: 31 December 2018). Vitamin B12 was given during breastfeeding via a nipple shield during a single breastfeed. A qualified nurse or lactation consultant known to the mother was available for breastfeeding support and advised on the appropriate application of the nipple shield to the breast. Assessments included quantitative measurements of serum vitamin B12 levels at baseline and 6–8 hours after the interventional breastfeed, and a mixed methods assessment of maternal expectations, experience, and acceptability. The interviewer was a female researcher trained in line with the institution's guidelines for clinical research. Interviews were led by a single interviewer for consistency. Primary outcome measure was the detection of the change in vitamin B12 concentration in the infants' blood 6–8 hours after delivery from a nipple shield during breastfeeding. Secondary outcome measures was the qualitative assessment of impact on maternal expectation, experience and acceptability.

### Study population and participant recruitment

Breastfeeding mother-infant dyads were recruited through convenience sampling from the neonatal unit and postnatal transitional care wards: Potentially eligible mother-infant dyads were identified by the clinical team, following which mothers were approached by the research team. Eligibility included infants with a corrected age of <12 months, clinically well and assessed by the lactation team to be competent at breastfeeding. Exclusion criteria were hypersensitivity to the vitamin B12 tablet's ingredients, short bowel syndrome, and malabsorption. No restrictions related to prior B12 supplementation or potential interactions with human milk were considered, as the study was solely designed as a proof-of-principle. The total number of participants was based on guidelines for feasibility studies, and qualitative research, and in alignment with the objective of reaching information saturation as well as a power calculation for changes in B12 levels [6, 7].

## Nipple shield and vitamin formulation

Commercial ultrathin contact nipple shields (Medela, UK) of 16 mm (size S), 20 mm (size M), or 24 mm (size L) size, as recommended by the lactation support team, and following evidence from an earlier scoping exercise with mothers [1], were selected and worn by mothers during the study feed. These nipple shields are commonly used on the hospital's maternal and neonatal wards, made from transparent BPA- and taste-free soft silicone. A sublingual vitamin B12 tablet (1000 μg Methylcobalamin, JustVitamins Ltd, UK), suitable for vegans and without known allergenic components, was placed into the shield's teat immediately prior to the feed, and the mother then breastfed as usual.

The tablet characteristics comprised the following: round tablet with a diameter of 8 mm and an average weight of 0.26 g; ingredients: Sorbitol, Stearic Acid, Beetroot, Blackcurrant, Magnesium Stearate, Methylcobalamin. To investigate the vitamin B12 tablets' disintegration properties, lab-based studies were performed. Hereby, the vitamin B12 tablets were placed in the silicone teat of a 20 mm Medela ultrathin contact nipple shield, and delivery into full-fat cow's milk investigated in triplicate using a breastfeeding simulation apparatus, capable of simulating both the process of lactation and infant feeding by mimicking the average flow rates/patterns of milk during breastfeeding and the infant's suckling pressure at physiological relevant conditions. The percentage of tablet released within 5, 10 and 20 min was approximately 41%, 62%, and 74% respectively, and no tablet break-offs during disintegration or dislocation of the remaining tablet through the silicone teat's three holes following the experiment was observed.

## Vitamin B12 levels

A baseline blood sample (venous blood or heal brick) was taken within a week prior to the study feed and a peak level at 6–8 hours after the study feed. The timing for collection was based on the pharmacokinetic profile and anticipated peak B12 levels [8]. Samples were immediately centrifuged and separated, with serum stored at -20˚C for later batch analysis. Serum vitamin B12 levels were analyzed using a LOCI vitamin B12 assay (Siemens Healthcare) at the Core Biochemical Assay Laboratory (CBAL), Cambridge University Hospitals.

## Mixed methods approach

A combination of tablet-based Likert-scale questionnaires and recorded semi-structured interviews, developed based on established guidelines [9], were undertaken before and after the study feed. Interviews were performed in a quiet area at the infants' cot side to avoid separation of mother and infants. Discussions focused on the evaluation of maternal expectation, experience, and acceptability of the intervention. Interviews were voice-recorded, but to support honest critical reporting, mothers also provided scored answers on a tablet device. Data on those scores was quantitatively evaluated, while semi-structured interviews were transcribed verbatim, and potentially identifiable data anonymized. Analysis was facilitated by ATLAS.ti (Scientific Software Development GmbH), using an inductive approach of thematic content analysis [10, 11]. Hereby, an initial coding framework emerged following pre-reading, a line-by-line open-coding approach, and regrouping steps. A final coding framework was developed, through iterative revisions within the research team.

## Statistical analyses and power calculation

The sample size was based on the literature for preterm infants' supplementation, in which B12 between 8–12 weeks led to an approximate three-fold rise in serum B12 levels. As such

this sample size would provide 90% power at the 5% level to demonstrate a two-fold increase in B12 [12–14] (two sided superiority). All data shown is presented as mean percentage or mean with standard deviations and ranges if normally distributed. Differences in serum B12 levels before and after the test-feed were assessed using paired student t-test. Exploratory analyses were undertaken in relation to impact of gestational age at birth and postnatal age on increases in vitamin B12 levels post the test feed. Analyses was undertaken using SPSS 26 and $p < 0.05$ was considered statistically significant.

## Ethics approval

The study was approved by the London, Brighton & Sussex Research Ethics Committee (18/LO/0551). All participants provided their written informed consent to take part and to be quoted anonymously in this publication. ClinicalTrials.gov Identifier: NCT03799367. https://clinicaltrials.gov/ct2/show/NCT03799367. The study was registered and released on 8 July 2018, prior to the start of patient recruitment. It was released to the public following review by the PRS protocol team on 10 January 2019. The authors confirm that there are currently no ongoing and related trials for this drug/intervention and that all existing ones were registered. Mother-infant dyads were recruited from 9 July until the end of December 2019. Following completion of the post-feed interview, enrolment in the study for a respective mother-infant dyad was completed.

## Results

The study aimed to recruit a maximum of 30 mother-infant dyads. A total of 43 mothers were approached, of which 26 (60%) consented to participation and 20 completed the full study protocol (Fig 1, Table 1). Reasons for non-completion/exclusion from analysis in the six cases were: a change to bottle feeding [1], discharge before study feed [2], lack of appropriate blood sample [2], and parent withdrawal [1]. All infants who started the test feed with the B12 completed the feeding intervention.

### Vitamin B12 delivery

The mean duration of the study feeds (n = 20) was 19.8 ± 5.5 min (Range 15–30 min). The lactation team and mothers did not report any aversive impact of the tablet's presence in the nipple shield on the process of feeding. Complete tablet disintegrating was achieved in all study feeds, with no residue tablet visible to the naked eye. Changes in serum B12 levels are provided in Table 2. Vitamin B12 levels rose in all infants from a mean baseline (range) 533 ± 188 pg/mL (236–925 pg/mL) to 1871 ± 1228 pg/mL (610–4981 pg/mL) at 6–8 hours post-delivery.

### Mixed methods assessment

The mean (range) duration of the semi-structured maternal interviews to assess expectations (pre-study feed) and experiences (post study feed) were 7.7 (4.4–16.6) minutes and 7.0 (3.5–12.2) minutes, respectively.

**Maternal expectations (pre-study feed).** Identified themes related to the uncertainty of risks but also curiosity for the intervention, as well as strong expectations of perceived emotional and practical benefits. Areas of concerns included the use of a nipple shield itself, and the therapeutic it contained. In particular, for mothers who had not previously used nipple shields, unknowns about the infant's reaction and behavior during the study feed were the predominant source of concern, including the infant's ability to latch and the alteration of normal breastfeeding behavior. "It might take some [time] getting used to. I don't know how she is

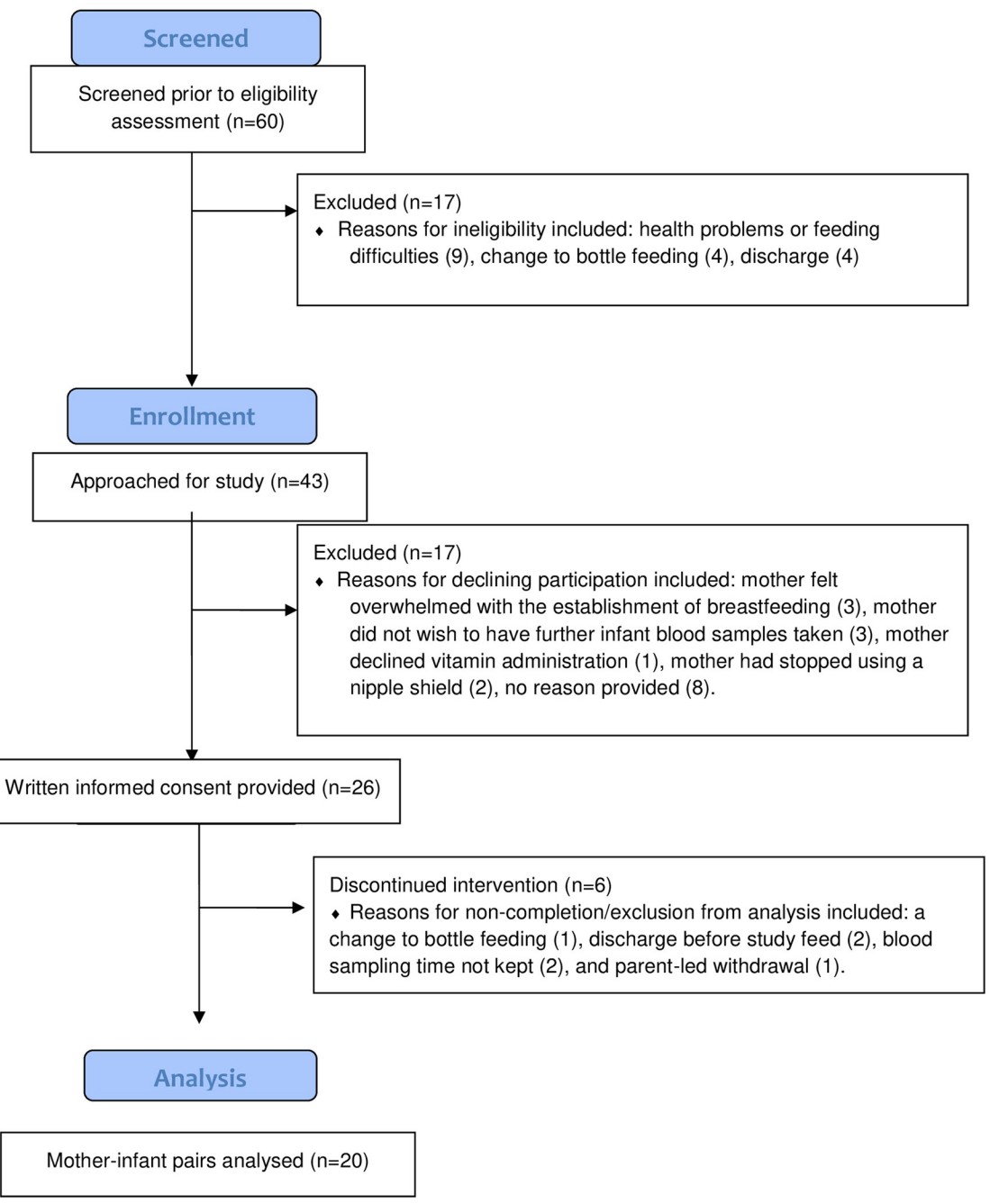

**Fig 1. Consort diagram.**

going to do with a nipple shield. I mean the way that she latches." (M12) In contrast, mother with previous experience using nipple shields, reflected exclusively on the tablet's disintegration properties, its potential impact on the milk's taste and implication on breastfeeding

**Table 1. Participant characteristics (mother-infant pairs, n = 20).**

| Characteristics | Mean (range) or N (%) |
|---|---|
| **Mothers' characteristics** | |
| Mother's age, mean (range) [years] | 32.4 (Range 23–39) |
| Total number of children, N (%) | |
| 1 | 9 (45) |
| 2–3 | 10 (50) |
| >3 | 1 (5) |
| **Infants' characteristics** | |
| Gestational age at birth, n (%) | |
| Preterm <32 weeks | 3 (15) |
| Preterm 32 to <37 weeks | 4 (20) |
| Term > 37 (37 to <41 weeks) | 13 (65) |
| Birth weight, mean (range) [gram] | 2769 (890–4145) |
| Age at time of study, mean (range) [days] | 16.2 (2–70) |
| Corrected gestational age at time of study, mean (range) [days] | -3.7 (-30-15) |
| Stay of infant on Neonatal Intensive Care, N (%) | |
| Yes, up to 1 week | 5 (25) |
| Yes, 1 week or longer | 7 (35) |
| No | 8 (40) |
| Exclusive breastfeeding at time of study, N (%) | |
| Yes | 7 (35) |
| No, also NG | 9 (45) |
| No, also bottle | 4 (20) |
| Use of nipple shield, N (%) | |
| For current infant | 9 (45) |
| Only for a previous infant | 1 (5) |
| Never | 10 (50) |

practice. "[. . .] will it dissolve, and will she. . . be able to taste it?" (M2)". . .if I was going to have a worry it would be that it would give them a negative experience of breastfeeding and then would put them off breastfeeding." (M6) Despite associated worries, mothers described their positivity and curiosity in attempting vitamin delivery during breastfeeding. "I think it's worth looking into. [. . .] I think it is a good idea." (M12) "It is just quite exciting to see how it works." (M2)

Participants expected an emotional improvement or an increased convenience of infant therapeutic delivery whilst feeding (Table 3). This included a reduction of stress for both mother and infant, or a perceived enhancement in physical intimacy, alongside a perception of this method as being more 'natural' (Table 3). Suggested potential practical benefits included time saving, fewer dosing errors and being "less messy". "It seems a more natural way of administering medication." (M5)

**Maternal experiences of the study feed (post feed).** A positive experience was reported by the majority of mothers along with a feeling of surprise in the ease of use of the nipple shields, and the lack of impact on the infant's breastfeeding behavior. Mothers attributed their surprise about their infants' contentment to the lack of experience and emotional factors relating to preconceptions about nipple shields. "What is this going to feel like? And is it going to be a barrier to feeding? And is he going to latch properly.' But actually, all of that was fine." (M4) Practical concerns remained regarding the potential of incomplete therapeutic delivery. "What would you do—if it was an actual drug–and [you had] given only part of a dose? (M2)

**Table 2. Changes in serum vitamin B12 levels from baseline to 6–8 hours after the study feed.**

| Infant ID | Gestational age at birth [weeks+days] | Age at time of study [days from birth] | Baseline serum vitamin B12 level (pre study feed) [pg/mL] | Peak serum vitamin B12 level (6–8 hours post study feed) [pg/mL] | Percentage change |
|---|---|---|---|---|---|
| 1 | 31+5 | 55 | 575 | 3484 | 506 |
| 2 | 26+5 | 70 | 681 | 2577 | 278 |
| 3 | 37+0 | 6 | 449 | 1743 | 288 |
| 4 | 37+2 | 6 | 858 | 1285 | 50 |
| 5 | 41+1 | 4 | 303 | 1045 | 245 |
| 6 | 31+3 | 30 | 236 | 4981 | 2011 |
| 7 | 38+4 | 21 | 593 | 1928 | 225 |
| 8 | 34+6 | 19 | 596 | 4104 | 589 |
| 9 | 41+1 | 6 | 430 | 1006 | 134 |
| 10 | 40+6 | 6 | 925 | 1121 | 21 |
| 11 | 37+6 | 4 | 565 | 1321 | 134 |
| 12 | 41+2 | 6 | 325 | 610 | 88 |
| 13 | 40+2 | 4 | 660 | 1259 | 91 |
| 14 | 40+2 | 6 | 582 | 1104 | 90 |
| 15 | 41+0 | 8 | 397 | 866 | 118 |
| 16 | 32+1 | 29 | 352 | 1506 | 328 |

Vitamin B12 levels of four infants were excluded as samples clearly showed haemolysis. Pg: pictogram, mL: millilitre.

**Overarching themes: Perceived advantages and acceptability.** Mothers emphasized the potential of this therapeutic delivery method to de-medicalize infant treatment and described it as "less invasive" (M7), and "not an aggressive method of delivery" (M16), "you are not forcing them" (M20). This appeared to relate to an emotional burden for mothers of infants with prior neonatal intensive care experience. "[. . .] for babies who've had to undergo all that, to have something so natural, is lovely." *(moved to tears)* (M12) "[. . .] I would like something more natural for her. [. . .] Everything that has to do with syringes and medication makes me think of NICU and, you know, this very scary part of her life." (M19) All mothers advocated for the availability of oral infant therapeutic administration during breastfeeding, with the majority preferring this method of infant therapeutic delivery (see Table 4), emphasizing that it would provide "choices" (M8).

**Table 3. Comparison of maternal expectations before and experiences reported after the study feed.**

| | Likert scale before study feed | Likert scale after study feed | Change [%] |
|---|---|---|---|
| The nipple shield with a vitamin tablet. . . | | | |
| . . .will be/was easier than using an oral syringe. | 7.0 ± 1.6 | 8.3 ± 1.8 | +19 |
| . . .will make/ made me less worried. | 7.2 ± 2.0 | 8.6 ± 1.5 | +19 |
| . . .will make/ made my baby feel less upset/ distressed. | 7.7 ± 1.5 | 8.6 ± 1.4 | +12 |
| . . .will help/ helped me to feel closer to my baby. | 7.7 ± 1.6 | 8.4 ± 1.7 | +9 |

The Likert scale was used (10 = highly agree, 0 = highly disagree). Numerical values are shown as mean ± standard deviation. Further quotes can be found in S1 Table.

**Table 4. Summary of reported maternal experience and acceptability of the study feed.**

| | Agreement [%] |
|---|---|
| **Experience** | |
| The nipple shield with a vitamin tablet. . . | |
| . . .was a positive experience | 95 |
| . . .was easy to use. | 95 |
| . . .was comfortable to wear. | 95 |
| My baby. . .. | |
| . . .latched as usual. | 95 |
| . . .breastfed as usual. | 90 |
| **Acceptability** | |
| I prefer to give medicines/ nutrients using a nipple shield over using an oral syringe. | 85 |
| I think the nipple shield could be an acceptable method for nutrient delivery. | 100 |
| I think the nipple shield could be an acceptable method for medicine delivery. | 95 |
| I would like that medicine/ nutrient delivery during breastfeeding becomes possible for parents in the future | 100 |

Further quotes can be found in S2 and S3 Tables.

## Discussion

This is the first clinical study to show that a solid formulation can be delivered successfully to infants during breastfeeding. The study demonstrated that the formulation used was easily dispersed in human milk during a breastfeed, with all infants showing a clinically significant rise in B12 serum levels. Neither mothers themselves, nor the lactation team, reported any apparent impact of the tablet's presence in the nipple shield on the process of feeding.

This feasibility study used B12 as a marker of utility for this novel delivery approach, not as a therapeutic option or to explore pharmacokinetic uptake in infants. However, the findings do highlight the complexities of bioavailability in the newborn not related only to breastfeeding, but also to gut pathology, postnatal age, and gestational age. Findings of this study suggest that vitamin B12 absorption was dependent on postnatal age and independent of prematurity or birth weight (total absorption: $0.0072 \pm 0.0029\%$ for infants $<7$ days of age, and $0.329 \pm 0.0173\%$ infants aged $19 - 70$ days). Given the small sample size, no conclusive statement can be made, about the interdependency between postnatal age and vitamin B12 absorption, and further investigations would be required. The findings are however in keeping with the literature for preterm infants' supplementation, in which B12 between 8–12 weeks led to an approximate three-fold rise in serum B12 levels [13, 14]." but the findings are in keeping with the literature on B12. Robust pharmacokinetic studies would be required for specific drugs to determine dosing and safety profile of individual preparations.

Previous studies have explored the putative benefits of solid formulations during breastfeeding in low resource settings [15]. A variety of compounds and drugs have been studied using an *in-vitro* model including zinc [3, 16]. Such delivery of zinc could help in the prevention and treatment of diarrheal disease and pneumonia which are significant causes of child mortality and morbidity [16], particularly since manufacturing costs for unbranded nipple shields are considered compatible to oral syringes and applicable for low and middle income countries. Others have explored the acceptability for administering antiviral agents to prevent mother-to-child transmission of HIV through breastfeeding [17]. Our research now is important as the first *in-vivo* study with mothers and infants, and therefore is a step change in realizing this potential.

Undertaking this study within a Neonatal referral service in the UK has also highlighted the potential of such an intervention to support breastfeeding mothers and their infants in this high-resource setting. Oral syringes, although often considered as non-invasive by the clinical team, evoked significant negative maternal responses. In contrast, despite some pre-conceptions about the use of nipple shields, none of the mothers reported any difficulties with their use. In addition, since past literature has shown that ultra-thin contact nipple shields have the potential to increase milk transfer for preterm infants and to prolong the duration of mothers breastfeeding, use even for young infants can be considered beneficial [18–20]. However, this study focused on infants prior to discharge home, and although nipple shields can be used with older infants we cannot comment on the impact on drug delivery to infants over 6 months of age.

Some of the mothers in this study had experienced the stresses of caring for an infant in intensive care. For these mothers, it was apparent that they perceived this study intervention as a more 'natural' method of drug delivery and a potential opportunity for further establishing maternal infant bonding, and for them to take back ownership. This is an important part of building confidence in parents ability to take responsibility for their infant's care [21–25]. Further studies would be important to understand the emotional impact of such an intervention in a wider cultural context. Ultimately, use of the nipple shield device for medication delivery would require either the usage of existing dosage forms that are shown to reliably disintegrate in breast milk within the duration of one breastfeed or the focused development of novel dosage forms within hospital facilities under CE-certified process or industrial partners. Provisional discussions with MHRA suggest that drug delivery in this way for regulatory purposes would be classified as a combination product.

None of the infants in this study showed baseline vitamin B12 deficiency, but all showed a significant increase in levels from baseline following a single interventional feed. In contrast, vitamin B12 deficiency is common in many low- and middle-income countries and in vegan mothers. Severe vitamin B12 deficiency is harmful to the developing infant brain, subclinical vitamin deficiency has been associated with decreased cognitive performance and supplementation can be beneficial [26]. The increase in levels did not show any relation to baseline levels or gestational age, and further studies would be needed on individual drugs to explore pharmacodynamics of this route of drug delivery and implications for dosing.

This is a single center study with a limited sample size, intentionally chosen to enable proof-of-concept assessment while reducing unnecessary infant exposure. Only mothers willing to use a nipple shield provided their consent, and therefore the sample may be biased towards a more favorable evaluation. However, approximately half the participants had not used a nipple shield previously, and only 5% of mothers declined participation due to concerns about its use. As each infant only experienced one interventional feed, the impact of repeated drug delivery in this way on latch would need further investigation. In this study, tablet taste did not appear to impact on feeding, and complete tablet delivery was achieved, which may reflect the infant's acceptance of a variety of tastes and that milk may have taste masking properties [27–30]. However, individual formulations would require testing to determine the impact of taste or texture during breastfeeding, and formulation adjustments as needed. While many paediatric formulations include taste masking techniques, excipients used in the neonatal population should be limited. 'However, it would be important to determine the potential impact of taste, as the bitter taste of some medicines might cause aversion to breast feeding, and adjustment to formulations would be required to insure compliance. There may be benefits from the nipple shield delivery if breastmilk is proven to provide taste masking effects".

## Conclusion

As the first *in-vivo* study, this study demonstrate a potential step change in drug delivery in breastfeeding infants. Along with demonstrating practical feasibility, this study has highlighted the importance of maternal preferences and potential impact on infant bonding. However, detailed further studies are needed to investigate the potential range of therapeutic formulations and to explore practical therapeutic uses.

## Supporting information

**S1 Checklist. TREND statement checklist.**
(PDF)

**S1 Table. Comparison of maternal expectations before and experiences after vitamin delivery during breastfeeding using a commercial nipple shield.**
(DOCX)

**S2 Table. Summary of maternal experience and assessment of breastfeeding during vitamin B12 delivery via a nipple shield whilst feeding.**
(DOCX)

**S3 Table. Summary of maternal acceptability of medicine and nutrient delivery during breastfeeding using a commercially available ultrathin contact nipple shield whilst feeding.**
(DOCX)

**S1 File.**
(DOC)

**S2 File. COREQ (consolidated criteria for reporting qualitative research) checklist.**
(DOCX)

## Acknowledgments

The authors like to thank the participants for providing their time to the study. For their support in facilitating the investigation, thanks is given to the nursing and medical teams on both the Neonatal Intensive Care Unit and the Transitional Care Unit at the University of Cambridge Addenbrooke's Hospital Trust. We would like to thank the Core Biochemical Assay Laboratory (CBAL), Cambridge University Hospitals, for their support in blood sample analysis, the National Institute for Health Research (NIHR) Cambridge Biomedical Research Centre, and colleagues at the University of Nottingham Medical School for their thought-partnership.

## Author Contributions

**Conceptualization:** Theresa Maier, Paula Peirce, Sophie L. Whitehouse, Nigel K. H. Slater, Kathryn Beardsall.

**Data curation:** Theresa Maier, Kathryn Beardsall.

**Formal analysis:** Theresa Maier, Kathryn Beardsall.

**Funding acquisition:** Theresa Maier, Kathryn Beardsall.

**Investigation:** Theresa Maier, Paula Peirce, Laura Baird, Kathryn Beardsall.

**Methodology:** Theresa Maier, Paula Peirce, Laura Baird, Sophie L. Whitehouse, Kathryn Beardsall.

**Project administration:** Theresa Maier, Kathryn Beardsall.

**Resources:** Theresa Maier.

**Software:** Theresa Maier.

**Supervision:** Nigel K. H. Slater, Kathryn Beardsall.

**Validation:** Theresa Maier.

**Visualization:** Theresa Maier.

**Writing – original draft:** Theresa Maier, Sophie L. Whitehouse, Kathryn Beardsall.

**Writing – review & editing:** Theresa Maier, Paula Peirce, Laura Baird, Sophie L. Whitehouse, Nigel K. H. Slater, Kathryn Beardsall.

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
