## [Decision Letter · Decision Letter 0]

10 May 2021

PONE-D-20-38364

Drug delivery from a solid formulation during breastfeeding – a clinical feasibility study with mothers and babies

PLOS ONE

Dear Dr. Beardsall,

Thank you for submitting your manuscript to PLOS ONE. After careful consideration, we feel that it has merit but does not fully meet PLOS ONE’s publication criteria as it currently stands. Therefore, we invite you to submit a revised version of the manuscript that addresses the points raised during the review process.

We look forward to receiving your revised manuscript.

Kind regards,

Nancy Beam, PhD

Staff Editor

PLOS ONE

Journal Requirements:

2)  Thank you for submitting your clinical trial to PLOS ONE and for providing the name of the registry and the registration number. The information in the registry entry suggests that your trial was registered after patient recruitment began. PLOS ONE strongly encourages authors to register all trials before recruiting the first participant in a study.

i) your reasons for your delay in registering this study (after enrolment of participants started);

ii) confirmation that all related trials are registered by stating: “The authors confirm that all ongoing and related trials for this drug/intervention are registered”.

3) In the Methods section of the manuscript please address the following:

a) Please specify when the baseline blood samples of the infants were taken as according to the protocol

b) Please clearly state the primary and secondary outcomes of the study

c) Please specify the date ranges over which you have recruited the participants

d) When reporting the results of qualitative research, we suggest consulting the COREQ guidelines: http://intqhc.oxfordjournals.org/content/19/6/349. In this case, please consider including more information on the number of interviewers, their training and characteristics; and please provide the interview guide used.

4) There is a discrepancy on the total number of participants recruited for the study described in the manuscript text and the online registry. Please provide some further clarification for this. Furthermore, it is stated that the mothers will participate in a pre-intervention interview in the protocol, however this is not stated in the manuscript text. Please provide this information in the body of the manuscript.

5) PLOS requires an ORCID iD for the corresponding author in Editorial Manager on papers submitted after December 6th, 2016. Please ensure that you have an ORCID iD and that it is validated in Editorial Manager. To do this, go to ‘Update my Information’ (in the upper left-hand corner of the main menu), and click on the Fetch/Validate link next to the ORCID field. This will take you to the ORCID site and allow you to create a new iD or authenticate a pre-existing iD in Editorial Manager. Please see the following video for instructions on linking an ORCID iD to your Editorial Manager account: https://www.youtube.com/watch?v=_xcclfuvtxQ

6) Please amend either the title on the online submission form (via Edit Submission) or the title in the manuscript so that they are identical.

7) Please provide a caption for Figure 1.

8) Please ensure that you refer to Figure 1 in your text as, if accepted, production will need this reference to link the reader to the figure.

9) Please include captions for your Supporting Information files at the end of your manuscript, and update any in-text citations to match accordingly. Please see our Supporting Information guidelines for more information: http://journals.plos.org/plosone/s/supporting-information.

10) We noted in your submission details that a portion of your manuscript may have been presented or published elsewhere. part of an oral presentation at the European Society for Developmental Perinatal and Paediatric Pharmacology Congress (doi: http://dx.doi.org/10.1136/archdischild-2019-esdppp.18 )

Please clarify whether this conference proceeding  was peer-reviewed and formally published. If this work was previously peer-reviewed and published, in the cover letter please provide the reason that this work does not constitute dual publication and should be included in the current manuscript.

Reviewers' comments:

Reviewer's Responses to Questions

**Comments to the Author**

1. Is the manuscript technically sound, and do the data support the conclusions?

Reviewer #1: No

Reviewer #2: Partly

2. Has the statistical analysis been performed appropriately and rigorously? 

Reviewer #1: No

Reviewer #2: No

3. Have the authors made all data underlying the findings in their manuscript fully available?

Reviewer #1: Yes

Reviewer #2: Yes

4. Is the manuscript presented in an intelligible fashion and written in standard English?

Reviewer #1: Yes

Reviewer #2: Yes

5. Review Comments to the Author

Reviewer #1: A clinical feasibility study was conducted which aimed to explore the potential to deliver medicine to infants via a solid formulation during breastfeeding in n=43 infant mother dyads. The delivery of Vitamin B12 through a solid formulation that dissolved in human breast milk did not impair breastfeeding. Furthermore, Vitamin B12 levels increased in all infants at 6-8 hours post-delivery. Eighty-five percent of mothers preferred the solid formation over syringe delivery.

Major revisions:

1- Include a statistical analysis section which summarizes all the statistical methods and cites the statistical software used for the analysis.

2- State and justify the study’s target sample size with a pre-study statistical power calculation.

Minor revisions:

1- In addition to summarizing the mean and range for continuous variables, provide standard deviations.

2- Table 3: Consider presenting these summary results in a graphic format. Specify the type of summary statistics presented.

3- Indicate the date range subjects were included in the study.

Reviewer #2: The study is the first to perform in vivo investigation in using nipple shield for drug delivery to infants and provides useful insight in the use. However, there are limitations to the study which require further consideration.

Minor points:

1. Line 21-23, please specify drug delivery to infants.

2. Line 63-64, specify blood of the infant.

3. Please check position of reference citation in the text. It should be before the full stop of the sentences.

Major points:

1. Please provide more information on the VB12 tablet: dimension, ingredients, total weight of the tablet, in vitro properties (disintegration, dissolution).

2. Please provide further description on the nipple shield, any pictures after the tablet is fitted? Whether the fitting of the tablet related to the tablet size, e.g. tablet too small might fall?

3. The major issue of the study is the huge variation of the VB12 serum concentration. The % increase differs massively (nearly 100 times the highest to the lowest). What would be the reason of this, if the dose was completed for every case as observed? Is this because of dose accuracy or absorption variation? This would be unacceptable to deliver medicines. Have the authors compared their data with previous literature on VB12 PK in paediatrics? This needs to be discussed and the limitation clearly stated.

4. The authors have mentioned taste issues. it is a good point that milk might mask the bitter taste of medicines. To which extend? How the medicines taste and dose affect this? What would be the taste masking strategy available? Need more discussion.

5. How long were the breadfeeding last? This would affect the disintegration and dissolution of the tablet.

6. Would the use of the shield become extemporaneous preparation of the medicine? what would be the regulatory indications?

7. The infants in this study are very young. Would older children (e.g. over 6 months) be more aware of the shield and impacted by the delivery?

8. The authors mentioned the use in LMIC. What is the implications on cost of the shield? Would the formulation needs to be specially designed and developed (a sublingual tablet is available for this API but might not be available for others) and would this increase cost?

9. It is true that this is a proof-of-concept study but the conclusion was quite strong in recommending this type of delivery. The authors need to have more careful and balanced evaluation of the technology to be able to provide this type of recommendation/conclusion. Otherwise, it could be misleading if it was taken out of the context.

6. PLOS authors have the option to publish the peer review history of their article (what does this mean?). If published, this will include your full peer review and any attached files.

Reviewer #1: No

Reviewer #2: No

---

## [Author Response · Author response to Decision Letter 0]

15 Jun 2021

15th June 2021

Re: Manuscript Number: PONE-D-20-38364

Drug delivery from a solid formulation during breastfeeding – a feasibility study with mothers and infants

Dear Ms Beam,

Many thanks for your kind response with regard to the article entitled “Drug delivery from a solid formulation during breastfeeding – a clinical feasibility study with mothers and infants” for publication in PLOS ONE. We have carefully considered the reviewers’ comments, and much appreciate their suggestions. We have made the suggested changes and provide details of further information / explanations listed below. 

All the changes made to the manuscript are indicated via tracked changes. A detailed response to the reviewers’ comments can be found on the following page.

We hope that the changes made will be to your and the reviewers’ complete satisfaction, but do let us know if you require any further information.

Sincerely,

Dr Kathryn Beardsall

 

Detailed response to the editor’s and reviewers’ comments

Editor

Editor’s comment 1: Please ensure that your manuscript meets PLOS ONE's style requirements, including those for file naming. 

Corresponding author’s response 1: The manuscript was alignment with the PLOS ONE style requirements.

Editor’s comment 2: Thank you for submitting your clinical trial to PLOS ONE and for providing the name of the registry and the registration number. The information in the registry entry suggests that your trial was registered after patient recruitment began. PLOS ONE strongly encourages authors to register all trials before recruiting the first participant in a study. As per the journal’s editorial policy, please include in the Methods section of your paper: i) your reasons for your delay in registering this study (after enrolment of participants started); ii) confirmation that all related trials are registered by stating: “The authors confirm that all ongoing and related trials for this drug/intervention are registered”.

Corresponding author’s response 2: The following statement has now been included in the method section of the revised manuscript: “The study was registered and released on 8 July 2018, prior to the start of patient recruitment. It was released to the public following review by the PRS protocol team on 10 January 2019. The authors confirm that there are currently no ongoing and related trials for this drug/intervention and that all existing ones were registered.” Proof of the first submission date of 8 July 2018 can be found on https://clinicaltrials.gov/ct2/show/record/NCT03799367 when clicking on the tab “Tabular view”, under the heading “First Submitted Date”.

Editor’s comment 3: In the Methods section of the manuscript please address the following: a) Please specify when the baseline blood samples of the infants were taken as according to the protocol; b) Please clearly state the primary and secondary outcomes of the study; c) Please specify the date ranges over which you have recruited the participants; d) When reporting the results of qualitative research, we suggest consulting the COREQ guidelines: http://intqhc.oxfordjournals.org/content/19/6/349. In this case, please consider including more information on the number of interviewers, their training and characteristics; and please provide the interview guide used.

Corresponding author’s response 3: The following paragraphs were added to the manuscript (added information shown as underlined text):

a) “A baseline blood sample (full blood; heal brick) was taken within a week prior to the study feed and a peak level at 6 - 8 hours after the study feed.”

b) “Assessments included quantitative measurements of serum vitamin B12 levels at baseline and 6 - 8 hours after the interventional breastfeed, and a mixed methods assessment of maternal expectations, experience, and acceptability by a single investigator to provide for consistency.” and “Primary outcome measure is the detection of the change in vitamin B12 concentration in the infants' blood 6-8 hours after the vitamin B12 delivery from a nipple shield during breastfeeding during breastfeeding. Secondary outcome measures is the qualitative assessment of impact on maternal expectation, experience and acceptability.”

c) “Mother-infant dyads were recruited from 9 July until the end of December 2019. Following completion of the post-feed interview, enrolment in the study for a respective mother-infant dyad was completed.”

d) “The interviewer was a female researcher on the unit and trained in line with the institution’s guidelines for clinical research. Interviews were led by only one interviewer for consistency.”

Editor’s comment 4: There is a discrepancy on the total number of participants recruited for the study described in the manuscript text and the online registry. Please provide some further clarification for this. Furthermore, it is stated that the mothers will participate in a pre-intervention interview in the protocol, however this is not stated in the manuscript text. Please provide this information in the body of the manuscript..

Corresponding author’s response 4: The following text was added to the manuscript: “The study intended to recruit a maximum of 30 mother-infant dyads. A total of 43 mothers were approached, of which 26 consented to participation in the study.” 

Editor’s comment 5: PLOS requires an ORCID iD for the corresponding author in Editorial Manager on papers submitted after December 6th, 2016. Please ensure that you have an ORCID iD and that it is validated in Editorial Manager. To do this, go to ‘Update my Information’ (in the upper left-hand corner of the main menu), and click on the Fetch/Validate link next to the ORCID field. This will take you to the ORCID site and allow you to create a new iD or authenticate a pre-existing iD in Editorial Manager. Please see the following video for instructions on linking an ORCID iD to your Editorial Manager account: https://www.youtube.com/watch?v=_xcclfuvtxQ

Corresponding author’s response 5: As requested, the ORCID iD for the corresponding author has been linked to the Editorial Manager account.

Editor’s comment 6: Please amend either the title on the online submission form (via Edit Submission) or the title in the manuscript so that they are identical.

Corresponding author’s response 6: The title of the manuscript was amended to “Drug delivery from a solid formulation during breastfeeding – a clinical feasibility study with mothers and infants”.

Editor’s comment 7: Please provide a caption for Figure 1.

Corresponding author’s response 7: Figure 1 had been given a caption (see reference within “Results” section). 

Editor’s comment 8: Please ensure that you refer to Figure 1 in your text as, if accepted, production will need this reference to link the reader to the figure.

Corresponding author’s response 8: Figure 1 had been referenced in the text (see “Results” section).

Editor’s comment 9: Please include captions for your Supporting Information files at the end of your manuscript, and update any in-text citations to match accordingly. Please see our Supporting Information guidelines for more information: http://journals.plos.org/plosone/s/supporting-information.

Corresponding author’s response 9: Captions for all Supporting Information files were included at the end of the manuscript (shown below), and in-text citations updated accordingly.

Supporting Information

• Table A.1. Comparison of maternal expectations before and experiences after vitamin delivery during breastfeeding using a commercial nipple shield.

• Table A.2. Summary of maternal experience and assessment of breastfeeding during vitamin B12 delivery via a nipple shield whilst feeding.

• Table A.3. Summary of maternal acceptability of medicine and nutrient delivery during breastfeeding using a commercially available ultrathin contact nipple shield whilst feeding.

• COREQ (COnsolidated criteria for REporting Qualitative research) Checklist

Editor’s comment 10: We noted in your submission details that a portion of your manuscript may have been presented or published elsewhere. part of an oral presentation at the European Society for Developmental Perinatal and Paediatric Pharmacology Congress (doi: http://dx.doi.org/10.1136/archdischild-2019-esdppp.18). Please clarify whether this conference proceeding was peer-reviewed and formally published. If this work was previously peer-reviewed and published, in the cover letter please provide the reason that this work does not constitute dual publication and should be included in the current manuscript. 

Corresponding author’s response 10: Only an abstract of the research study was formally published to introduce the content of the oral presentation at the ESDPP congress – yet this abstract was not peer-reviewed. We confirm that the manuscript has not been peer-reviewed and published. 

Reviewer 1

Reviewer’s major point 1: Include a statistical analysis section which summarizes all the statistical methods and cites the statistical software used for the analysis.

Corresponding author’s response to major point 1: All data shown is presented as mean percentage or mean and ranges if normally distributed. The power calculation for the study has been added and an explanation that given the range of gestational and postnatal ages of the infants recruited exploratory analyses of these baseline characteristics was undertaken.

Reviewer’s major point 2: State and justify the study’s target sample size with a pre-study statistical power calculation.

Corresponding author’s response to major point 2: The following text was added to the manuscript: “The sample size was based on the literature for preterm infants’ supplementation, in which B12 between 8-12 weeks led to an approximate three-fold rise in serum B12 levels. As such this sample size would provide 90% power at the 5% level to demonstrate a 2 two-fold increase in B12.”

Reviewer’s minor point 1: In addition to summarizing the mean and range for continuous variables, provide standard deviations.

Corresponding author’s response to minor point 1: The following text was added to the manuscript: “Vitamin B12 levels rose in all infants from a mean baseline (range) 533 ± 188 pg/mL (236 - 925 pg/mL) to 1871 ± 1228 pg/mL (610 – 4981 pg/mL) at 6 - 8 hours post-delivery.”

Reviewer’s minor point 2: Table 3: Consider presenting these summary results in a graphic format. Specify the type of summary statistics presented.

Corresponding author’s response to minor point 2: The suggestion for a graphical illustration was carefully considered, yet a table format preferred. With regard to the summary statistics, the following text was added to the manuscript: “Numerical values are shown as mean ± standard deviation.”

Reviewer’s minor point 3: Indicate the date range subjects were included in the study.

Corresponding author’s response to minor point 3: The following text was added to the manuscript (added paragraphs / parts shown as underlined text): 

• “A baseline blood sample (venous blood or heal brick) was taken within a week prior to the study feed and a peak level at 6 - 8 hours after the study feed.” 

• “Mother-infant dyads were recruited from 9 July until the end of December 2019. Following completion of the post-feed interview, enrollment in the study for a respective mother-infant dyad was completed.” 

Reviewer 2

Reviewer’s major point 1: Please provide more information on the VB12 tablet: dimension, ingredients, total weight of the tablet, in vitro properties (disintegration, dissolution).

Corresponding author’s response to major point 1: The following text was added to the manuscript: “The tablet characteristics comprised the following: round tablet with a diameter of 8 mm and an average weight of 0.26 g; ingredients: Sorbitol, Stearic Acid, Beetroot, Blackcurrant, Magnesium Stearate, Methylcobalamin. To investigate the vitamin B12 tablets' disintegration properties, lab-based studies were performed. Hereby, the vitamin B12 tablets were placed in the silicone teat of a 20 mm Medela ultrathin contact nipple shield, and delivery into full-fat cow's milk investigated in triplicate using a breastfeeding simulation apparatus, capable of simulating both the process of lactation and infant feeding by mimicking the average flow rates/patterns of milk during breastfeeding and the infant’s suckling pressure at physiological relevant conditions. The percentage of tablet released within 5, 10 and 20 min was approximately 41%, 62%, and 74% respectively, and no tablet break-offs during disintegration or dislocation of the remaining tablet through the silicone teat's three holes following the experiment was observed.”

Reviewer’s major point 2: Please provide further description on the nipple shield, any pictures after the tablet is fitted? Whether the fitting of the tablet related to the tablet size, e.g. tablet too small might fall?

Corresponding author’s response to major point 2: The following text was added to the manuscript (added paragraphs / parts shown as underlined text): “Commercial ultrathin contact nipple shields (Medela, UK) of 16 mm (size S), 20 mm (size M), or 24 mm (size L) size, as recommended by the lactation support team, and following evidence from an earlier scoping exercise with mothers (1), were selected and worn by mothers during the study feed. These nipple shields are commonly used on the hospital’s maternal and neonatal wards, made from transparent BPA- and taste-free soft silicone.”

Reviewer’s major point 3: The major issue of the study is the huge variation of the VB12 serum concentration. The % increase differs massively (nearly 100 times the highest to the lowest). What would be the reason of this, if the dose was completed for every case as observed? Is this because of dose accuracy or absorption variation? This would be unacceptable to deliver medicines. Have the authors compared their data with previous literature on VB12 PK in paediatrics? This needs to be discussed and the limitation clearly stated.

Corresponding author’s response to major point 3: The following text was added to the manuscript: “This feasibility study used B12 only as a marker of utility for the novel delivery approach, not as a therapeutic option or to explore pharmacokinetic uptake in infants. However, the findings do raise an important observation of the complexities of bioavailability in the newborn not related only to breastfeeding, but also to gut pathology, postnatal age, and gestational age. Findings of this study seemed to suggest that vitamin B12 absorption was dependent on postnatal age and independent of prematurity or birth weight (total absorption: 0.0072 ± 0.0029 % for infants <7 days of age, and 0329 ± 0.0173 % infants aged 19 − 70 days). Given the small sample size, no conclusive statement can be made, and further investigations would be required but the findings are in keeping with the literature”

Reviewer’s major point 4: The authors have mentioned taste issues. it is a good point that milk might mask the bitter taste of medicines. To which extend? How the medicines taste and dose affect this? What would be the taste masking strategy available? Need more discussion.

Corresponding author’s response to major point 4: The following text was added to the manuscript (added paragraphs / parts shown as underlined text): “However, individual formulations would require testing to determine the impact of taste or texture during breastfeeding, and formulation adjustments as needed. While many formulations include taste advancements, masking ingredients are generally avoided in the neonatal population and there may be benefits from this route and the masking effects of milk on the taste of the medication.”

Reviewer’s major point 5: How long were the breastfeeding last? This would affect the disintegration and dissolution of the tablet.

Corresponding author’s response to major point 5: The following text was added to the manuscript: “The mean duration of the study feeds (n=20) was 19.8 ± 5.5 min (Range 15 – 30 min).”

Reviewer’s major point 6: Would the use of the shield become extemporaneous preparation of the medicine? what would be the regulatory indications?

Corresponding author’s response to major point 6: The following text was added to the manuscript: “Ultimately, use of the nipple shield device for medication delivery would require either the usage of existing dosage forms that are shown to reliably disintegrate in breast milk within the duration of one breastfeed or the focused development of novel dosage forms within hospital facilities under CE-certified process or industrial partners. Provisional discussions with MHRA suggest that drug delivery in this way for regulatory purposes would be classified as a combination product”

Reviewer’s major point 7: The infants in this study are very young. Would older children (e.g. over 6 months) be more aware of the shield and impacted by the delivery?

Corresponding author’s response to major point 7: The following text was added to the manuscript: “This study focused on infants prior to discharge home, and although nipple shields can be used with older infants we cannot comment on the impact on drug delivery to infants over 6 months of age.”

Reviewer’s major point 8: The authors mentioned the use in LMIC. What is the implications on cost of the shield? Would the formulation needs to be specially designed and developed (a sublingual tablet is available for this API but might not be available for others) and would this increase cost?

Corresponding author’s response to major point 8: The following text was added to the manuscript:

• “Such delivery of zinc could help in the prevention and treatment of diarrheal disease and pneumonia which are significant causes of child mortality and morbidity (13), particularly since manufacturing costs for unbranded nipple shields are considered compatible to oral syringes and applicable for low and middle income countries.” 

• “Ultimately, use of the nipple shield device for medication delivery would require either the usage of existing dosage forms that are shown to reliably disintegrate in breast milk within the duration of one breastfeed or the focused development of novel dosage forms within hospital facilities under CE-certified process or industrial partners.”

Reviewer’s major point 9: It is true that this is a proof-of-concept study but the conclusion was quite strong in recommending this type of delivery. The authors need to have more careful and balanced evaluation of the technology to be able to provide this type of recommendation/conclusion. Otherwise, it could be misleading if it was taken out of the context.

Corresponding author’s response to major point 9: The following text was added to the manuscript: “However, detailed further studies are needed to investigate the potential range of therapeutic formulations and to explore practical therapeutic uses.”

Reviewer’s minor point 1: Line 21-23, please specify drug delivery to infants.

Corresponding author’s response to minor point 1: The following addition was made to line 21-23 (see underlined text): “ In low-resource settings the delivery of medications, such as de-worming and retrovirals as well as nutrient and mineral supplementation, in liquid formulations is challenging, as liquids need to be refrigerated for storage, and syringes require clean water for sterilization”

Reviewer’s minor point 2: Line 63-64, specify blood of the infant.

Corresponding author’s response to minor point 2: The following addition was made to line 21-23 (see underlined text): “A baseline blood sample (full blood; heal brick) was taken prior to and a peak level at 6 - 8 hours after the study feed.”

Reviewer’s minor point 3: Please check position of reference citation in the text. It should be before the full stop of the sentences.

Corresponding author’s response to minor point 3: The position of reference citation in the text was reviewed and adjustments made in line with the requirements.

---

## [Decision Letter · Decision Letter 1]

6 Oct 2021

PONE-D-20-38364R1Drug delivery from a solid formulation during breastfeeding – a clinical feasibility study with mothers and infantsPLOS ONE

Dear Dr. Beardsall,

Thank you for submitting your manuscript to PLOS ONE. After careful consideration, we feel that it has merit but does not fully meet PLOS ONE’s publication criteria as it currently stands. Therefore, we invite you to submit a revised version of the manuscript that addresses the points raised during the review process.

The reviewers have identified some outstanding methodological and statistical clarifications that need to be provided before the submission can fully meet PLOS ONE's publication criteria.

We look forward to receiving your revised manuscript.

Kind regards,

Jamie Males

Staff Editor

PLOS ONE

Journal Requirements:

Reviewers' comments:

Reviewer's Responses to Questions

**Comments to the Author**

1. If the authors have adequately addressed your comments raised in a previous round of review and you feel that this manuscript is now acceptable for publication, you may indicate that here to bypass the “Comments to the Author” section, enter your conflict of interest statement in the “Confidential to Editor” section, and submit your "Accept" recommendation.

Reviewer #1: (No Response)

Reviewer #2: (No Response)

2. Is the manuscript technically sound, and do the data support the conclusions?

Reviewer #1: Yes

Reviewer #2: (No Response)

3. Has the statistical analysis been performed appropriately and rigorously? 

Reviewer #1: Yes

Reviewer #2: (No Response)

4. Have the authors made all data underlying the findings in their manuscript fully available?

Reviewer #1: Yes

Reviewer #2: (No Response)

5. Is the manuscript presented in an intelligible fashion and written in standard English?

Reviewer #1: Yes

Reviewer #2: (No Response)

6. Review Comments to the Author

Reviewer #1: Minor revision:

Line 114: Indicate the statistical testing method which achieves 90% power.

Reviewer #2: Thank you for addressing the comments and revising the manuscript.

In the marked up manuscript line 230-232, "Given the small sample size, no conclusive statement can be made,

and further investigations would be required but the findings are in keeping with the literature

on B12.", can you please specify "no conclusive statement can be made" on what? Also please specify e.g. "keeping with the literature on B12 absorption in infants" and provide literature references.

The authors have commented that oral syringe would need to be sterilised which is a disadvantage in LMICs. Would the nipple shield need to be sterilised before use?

This sentence needs to be written clearer "While many formulations

include taste advancements, masking ingredients are generally avoided in the neonatal

population and there may be benefits from this route and the masking effects of milk on the

taste of the medication." For example "While many paediatric formulations include taste masking techniques, excipients used in the neonatal population should be limited. There may be benefits from the nipple shield delivery if breastmilk is proven to provide taste masking effects."

Please also add a statement to the above sentence that the bitter taste of some medicines might cause aversion of infants to breastfeeding.

7. PLOS authors have the option to publish the peer review history of their article (what does this mean?). If published, this will include your full peer review and any attached files.

Reviewer #1: No

Reviewer #2: No

---

## [Author Response · Author response to Decision Letter 1]

24 Nov 2021

Re: Drug delivery from a solid formulation during breastfeeding – a feasibility study with mothers and infants

Manuscript Number: PONE-D-20-38364R1

Dear Jamie Males,

Many thanks for your helpful comments with regard to the full-length research article entitled “Drug delivery from a solid formulation during breastfeeding – a clinical feasibility study with mothers and babies” for publication in PLOS ONE. We have carefully considered the reviewers’ comments, and appreciate their suggestions. We have provided further information / explanations to paragraphs of the manuscript as enquired by you and the reviewers.

All the changes made to the manuscript are indicated via tracked changes. A detailed response to the reviewers’ comments can be found on the following page.

We hope that the changes made are to your and the reviewers’ complete satisfaction. 

Sincerely,

Dr. Kathryn Beardsall

 

Detailed response to the reviewers’ comments

In the following both the reviewers’ comments, as well as the actions taken are illustrated.

Reviewer 1

Minor point: Line 114 – Indicate the statistical testing method which achieves 90% power.

Response: Thank you we have added (two sided superiority) to clarify. 

Reviewer 2

 Minor point 1: In the marked up manuscript line 230-232, 

a) "Given the small sample size, no conclusive statement can be made, and further investigations would be required but the findings are in keeping with the literature on B12.", can you please specify "no conclusive statement can be made" on what? 

The text was amended to 

“Given the small sample size, no conclusive statement can be made about the interdependency between postnatal age and vitamin B12 absorption, and further investigations are required.”

b) Also please specify e.g. "keeping with the literature on B12 absorption in infants" and provide literature references.

References were added as requested: “This is a feasibility study but based on the literature for preterm infants’ supplementation, in which B12 between 8-12 weeks led to an approximate three-fold rise in serum B12 levels (9,10).”

Minor point 2: The authors have commented that oral syringe would need to be sterilised which is a disadvantage in LMICs. Would the nipple shield need to be sterilised before use?

Response: The initial suggestion was for a disposable or biodegradable nipple shield to avoid the need for sterilization but given that this was a proof of principle in a level 3 unit in the UK we thought this distracted from the main message of the paper. 

Minor point 3: This sentence needs to be written clearer "While many formulations include taste advancements, masking ingredients are generally avoided in the neonatal population and there may be benefits from this route and the masking effects of milk on the taste of the medication." For example "While many paediatric formulations include taste masking techniques, excipients used in the neonatal population should be limited. There may be benefits from the nipple shield delivery if breastmilk is proven to provide taste masking effects.” 

Response: Thank you we have amended as suggested

Minor point 4: Please also add a statement to the above sentence that the bitter taste of some medicines might cause aversion of infants to breastfeeding.

Response: The following sentence has been added. ‘However, it would be important to determine the potential impact of taste, as the bitter taste of some medicines might cause aversion to breast feeding, and adjustment to formulations would be required to insure compliance.’

---

## [Decision Letter · Decision Letter 2]

21 Jan 2022

PONE-D-20-38364R2Drug delivery from a solid formulation during breastfeeding – a clinical feasibility study with mothers and infantsPLOS ONE

Dear Dr. Beardsall,

Thank you for submitting your manuscript to PLOS ONE. After careful consideration, we feel that it has merit but does not fully meet PLOS ONE’s publication criteria as it currently stands. Therefore, we invite you to submit a revised version of the manuscript that addresses the points raised during the review process. The revised manuscript is close to being accepted for publication. However the authors have not directly addressed the issue regarding statistical test raised by reviewer #1. The authors have incorrectly responded that the test used was 2 sided superiority. What they were expected to mention was whether it was paired test for a before and after design and whether it was paired student t-test or Wilcoxon signed rank sum test or whatever. 

We look forward to receiving your revised manuscript.

Kind regards,

Sourabh Dutta

Academic Editor

PLOS ONE

Journal Requirements:

Reviewers' comments:

Reviewer's Responses to Questions

**Comments to the Author**

1. If the authors have adequately addressed your comments raised in a previous round of review and you feel that this manuscript is now acceptable for publication, you may indicate that here to bypass the “Comments to the Author” section, enter your conflict of interest statement in the “Confidential to Editor” section, and submit your "Accept" recommendation.

Reviewer #1: (No Response)

Reviewer #2: All comments have been addressed

2. Is the manuscript technically sound, and do the data support the conclusions?

Reviewer #1: Yes

Reviewer #2: Yes

3. Has the statistical analysis been performed appropriately and rigorously? 

Reviewer #1: Yes

Reviewer #2: Yes

4. Have the authors made all data underlying the findings in their manuscript fully available?

Reviewer #1: Yes

Reviewer #2: Yes

5. Is the manuscript presented in an intelligible fashion and written in standard English?

Reviewer #1: Yes

Reviewer #2: Yes

6. Review Comments to the Author

Reviewer #1: This prior comment was not adequately addressed:

Minor point: Line 115 – Indicate the statistical testing method which achieves 90% power.

Examples of statistical testing methods are t-tests, chi-square tests, etc.

Reviewer #2: Thank you to the authors to address all previous comments. The manuscript has been revised sufficiently to be published.

7. PLOS authors have the option to publish the peer review history of their article (what does this mean?). If published, this will include your full peer review and any attached files.

Reviewer #1: No

Reviewer #2: No

---

## [Author Response · Author response to Decision Letter 2]

24 Jan 2022

25th Jan 2022

Re: Drug delivery from a solid formulation during breastfeeding – a feasibility study with mothers and infants

Manuscript Number: PONE-D-20-38364R1

Dear Jamie Males,

Many thanks for your helpful comments with regard to the full-length research article entitled “Drug delivery from a solid formulation during breastfeeding – a clinical feasibility study with mothers and babies” for publication in PLOS ONE. We have addressed the outstanding request to include details of statistical methodology and this is highlighted in the manuscript. 

We hope that the changes made are to your complete satisfaction. Please let us know if you require any further information. 

Sincerely

Dr Kathryn Beardsall

---

## [Editor Report · Decision Letter 3]

17 Feb 2022

Drug delivery from a solid formulation during breastfeeding – a clinical feasibility study with mothers and infants

PONE-D-20-38364R3

Dear Dr. Beardsall,

We’re pleased to inform you that your manuscript has been judged scientifically suitable for publication and will be formally accepted for publication once it meets all outstanding technical requirements.

Kind regards,

Sourabh Dutta

Academic Editor

PLOS ONE
---

## [Editor Report · Acceptance letter]

24 Feb 2022

PONE-D-20-38364R3 

Drug delivery from a solid formulation during breastfeeding – a feasibility study with mothers and infants 

Dear Dr. Beardsall:

I'm pleased to inform you that your manuscript has been deemed suitable for publication in PLOS ONE. Congratulations! Your manuscript is now with our production department. 

Kind regards, 

on behalf of

Dr. Sourabh Dutta 

Academic Editor

PLOS ONE